# Fabrication of Enhanced UV Protective Cotton Fabric Using Activated Nano-Biocarbon Derived from Teff Hay Grafted by Polyaniline: RSM-Based Optimization and Characterization

**DOI:** 10.3390/molecules28135142

**Published:** 2023-06-30

**Authors:** Kibebe Sahile, Esayas Alemayehu, Abebe Worku, Sundramurthy Venkatesa Prabhu, Bernd Lennartz

**Affiliations:** 1Department of Chemical Engineering, College of Biological and Chemical Engineering, Addis Ababa Science and Technology University, Addis Ababa P.O. Box 16417, Ethiopia; kibebe.sahele@aastu.edu.et; 2Faculty of Civil and Environmental Engineering, Jimma University, Jimma P.O. Box 378, Ethiopia; esayas16@yahoo.com; 3Africa Center of Excellence for Water Management, Addis Ababa University, Addis Ababa P.O. Box 1176, Ethiopia; 4Department of Environmental Engineering, College of Biological and Chemical Engineering, Center of Excellence for Sustainable Energy, Addis Ababa Science and Technology University, Addis Ababa P.O. Box 16417, Ethiopia; abebe.worku@aastu.edu.et; 5Faculty of Agricultural and Environmental Sciences, University of Rostock, Justus-Von-Liebig-Weg6, 18059 Rostock, Germany

**Keywords:** nanocarbon, cotton fabric, ultraviolet resistance, biomass, FTIR, SEM, shielding properties

## Abstract

In the present study, a hybrid cotton fabric with an enhanced ultraviolet (UV) shielding property was developed by coating with functionally activated nanocarbon (FACN) which was grafted by polyaniline (PANI) using in situ polymerization. In light of this, Teff hay biomass was used to establish the activated nanocarbon (ANC), that was subsequently given a surface functionalization using a silane coupling agent. Using the response surface (RSM) statistical analysis, the study was optimized for the weight percent of ANC and PANI with respect to the cotton fabric that was found to offer remarkable UV protection, with an ultraviolet protection factor (UPF) of 64.563, roughly 17 times more than that of primitive cotton (UPF = 3.7). The different characterization techniques, such as UV absorption, Fourier transform infrared (FTIR), scanning electron microscopy (SEM), energy dispersive spectroscopy (EDS), and thermal behavior studies were investigated. In addition, the basic textile properties on optimized hybrid material were found to be appreciably increased. The results suggested that activated FACN made from Teff hay could be an effective alternative organic source material for developing UV protective hybrid cotton fabrics.

## 1. Introduction

Ternary composites are now being extensively studied since they been have proven to be more effective for improving the different crucial characteristics that must be met for different industrial demands [1]. In light of this, the development of improved ultraviolet (UV) protective fabric-based material seems to be more essential in terms of the fabric and can be used in extensive applications such as outdoor clothing, sports, automotive, and food packaging industries, etc. [2]. As such, cotton fabric has been widely used in the clothing industry. In addition to its different industrial applications, the wearable business is increasingly receiving attention within society; however, it has low resistance to UV rays, which are sensitive to microorganisms, and hygroscopic properties curb the use of cotton fabrics in a number of high-end industrial application areas as a potential functional material [3]. In this context, some of the most widely studied techniques, such as impregnating organic dyes, the application of hybrid polymers, and the coating of nanomaterials, have been observed to be more effective in improving the functional properties with respect to the UV protection of cotton fabrics [4]. Among these approaches, it has been proven that nanomaterials can be effectively exploited as functional materials to develop certain textile properties, specifically UV blocking, their antistatic, water-repelling, strength-enhancing, wrinkle resistance, and antibacterial abilities, in addition to their applications in wearable electronics, etc. [5]. The self-assembly coating of nanomaterials on the cotton fabric can effectively improve the UV blocking potential due to a large specific surface area using a very low concentration while being compared to its microparticle counterpart [6].

In other studies, a multifunctional cotton fiber with high electrical conduction and ultraviolet radiation fortification characteristics was effectively invented using the vacuum filtration deposition (VFD) technique to coat a graphite oxide (GO) nanosheet dispersal on the fabric surface, and then assembling the coated fabric with polyaniline (PANI) using an in situ chemical polymerization treatment [7]. Based on the photocatalytic results, it appears that the synergistic effects of PANI and TiO2 allow for PANI/TiO_2_/cotton composites to achieve an absorption and degradation efficiency of up to 87.67% when exposed to rhodamine B (RHB) [8]. The “in situ” polymerization of aniline (ANI) in a single-step procedure without re-doping has led to the creation of polyaniline-coated conductive textiles (woven textile material). In light of this, the greatest findings for outstanding UV resistance were achieved using different weight ratios of aniline and functionalized materials [9].

In this context, activated biocarbon (AC) can be an appropriate functionalized material that can be used to develop cotton-based hybrid composites. AC may be made from biomass residues, a resource that is both cheap and abundant [10]. In general, the thermo-chemical approach was used to convert agro-residues into activated carbons [11]. Ethiopia has a lot of agro-residual waste (hay) from Teff (*Eragrostis tef*), which might be used as the raw material for making things with added value, but it is not being put to good use [12]. In other studies, researchers have proved that multifunctional textiles can be made from cotton-based industrial textiles using the AC and ANI monomer [13] for enhancing UV resistance. Consequently, both PANI and nano-sized activated carbon (ANC) derived from Teff hay were employed in the current study.

Considering the above, the aim of this research was to develop cotton fabric-based nanocomposites using various concentrations of ANC and polyaniline to obtain an increased material quality by characterizing their UV-shielding property, chemical composition, and thermal performance [14]. Since the coating has a unique trap structure consisting of arrays of ANC and PANI, this work focused on the polymerization of the ANI monomer in the presence of ANC in varying weight ratios, all while vigorously and continuously stirring in an acidic medium to create unique polyaniline–ANC nanocomposites [15].

Furthermore, the usage of a weight ratio for ANI/ANC to develop the composite materials for enhancing the UV protection property was optimized using the response surface methodology (RSM) technique [16]. Then, in addition to the UV protection factor, various characteristics of the resulting hybrid-coated textile—which was developed using optimized composition of ANI and ANC—were also investigated.

## 2. Materials and Methods

Plain woven 100% cotton fabric with a weight per square meter of 96 g/m^2^, a warp yarn count of 44s Ne, and a weft yarn count of 40s Ne was procured from Bahir Dar Textile S. Co. (Bahir Dar, Ethiopia). The Teff hay used in this study was gathered in the Teff fields of Addis Ababa, Ethiopia. Merck, an Indian firm, supplied the aniline, ammonium persulphate (APS), ethanol, sodium hydroxide, triethoxysilane (3-APS), bromide (3-KBr), and acetic acid.

### 2.1. Preparation of ANC from Teff Hay

The ANC used in this study was made from Teff hay in line with the published literature [17]. The dry Teff hay was first carbonized at 500 °C in a nitrogen environment for 90 min, after being thoroughly washed with water to remove any impurities. The carbonized sample was then dried at 120 °C for 12 h after being digested with strong sodium hydroxide in a weight ratio of 1:3. To accomplish full graphitization, carbonization was performed at 400 °C for 30 min, followed by 800 °C for 60 min. The pH (6.0–7.0) was maintained in the carbonized product by neutralizing it with 0.1 M hydrochloride solution and washing it many times in distilled water. After 24 h of vacuum drying at 120 °C, the carbonaceous product sample was ready for ball milling to reduce its size to attain a nanostructured material followed by surface functionalization. Ball milling is one of the traditional strategies used to produce nanostructured materials. In this present study, a planetary centrifugal ball mill (TMAX-PBM-H, Xiamen Tmax Equipment, Xiamen, China) was used with stainless steel balls (diameter, 2.4 mm; number of balls, 800; total weight of the balls, 45 g). In order to prevent balls and biocarbon overheating, the ball milling experiment was set to operate with 5 min ON and 5 min OFF. After 14 h of milling operation, the average particle size and particle size distribution of the grounded biocarbon sample were carried out using a laser beam scattering instrument (Zetasizer Nano S90, Malvern Instruments, Malvern, UK). During the investigation of the particle size distribution, 1 mg of the activated biocarbon sample was dispersed in 200 mL of distilled water containing 0.5% Tween 80 and 1% ethanol using a magnetic stirrer for 50 min.

### 2.2. Surface Functionalization of ANC

In order to make the produced ANC compatible with aniline, the surface was functionalized with an amine group, as described in the published process [18]. As such, functional groups like acid, hydroxyl, etc. can be formed on the surfaces of the ANC using a three-hour heat treatment at 300 °C. The material was then homogenized in ethanol using ultrasonic agitation (UP100H, Sonics and Materials, Newtown, CT, USA) for 30 min before the addition of a measured dose of 3-APS. After 24 h of reflux with acetic acid to lower the pH to 4.5, the solids were strained out. Separation by centrifugation was followed by three washes of functionalized activated nanocarbon (FANC) in ethanol and then hexane to remove any unreacted silane, and finally, 24 h of drying in a vacuum oven set to 60 °C [19].

### 2.3. Fabrication of FANC-Coated and PANI-Grafted Cotton Fiber Samples

Based on the preliminary experiments using the one-variable-at-a-time (OVAT) approach, cotton fabrics with various combinations of aniline monomer and FANC (by weight % of cotton) were coated on cotton fabric and the UV-blocking property was determined [20,21,22,23]. Using the OVAT results, further optimization limits were determined for the concertation of the ANI monomer and FANC. As such, Table 1 lists the central composite design (CCD) combinations of percentages by weight for ANI monomer and FANC that were extracted from Design Expert software (Version 13, Stat-Ease, Inc. Minneapolis, MN, USA). The considered weight ratios were relative to the weightage of cotton fiber.

Figure 1 describes the schematic procedure for the in situ polymerization of activated carbon-doped polyaniline on cotton fabric. During the fabrication of composite development, sonication was used to break up the compound in 1 M HCl solution for an hour. At a temperature of 10 °C or below, the necessary amount of APS dissolved in 1 M HCl was applied over the course of 1 h to induce the full polymerization of aniline [24]. For an additional 3 h, at the same temperature, the mixture was sonicated. The FANC-coated polyaniline-grafted hybrid fabric that formed next was rinsed in 1 M HCl to dispose of any leftover excess oxidant and unreacted monomers. Furthermore, the hybrid material was given a bath in water and dried at 50 °C for 12 h. After grafting a polyaniline network onto a cotton surface, the resulting fabrics took on a dark green hue [25]. According to Table 1, the hybrid samples were prepared using different combinations of weight percentage of ANI and FANC (hereafter, the FACN-coated hybrid cotton sample is denoted by FANCPCF). Figure 2 represents a schematic depiction of the reaction mechanism for the grafting process whereby a hybrid nanocomposite was created.

### 2.4. Characterization Techniques

#### 2.4.1. Fourier Transform Infrared Spectroscopy (FTIR)

An FTIR measuring instrument (FTIR 7600, Shimadzu, Kyoto, Japan) was employed to collect the IR spectrum data. As a standard, KBr was applied. In order to record the spectrum, it was pelletized and mashed with samples (2%) [26].

#### 2.4.2. UV–Visible Spectroscopy

A spectrophotometer (UV 2450, Shimadzu, Japan) was used to record the UV–Visible transmittance spectra. The spectrum was captured using cloth samples that were 1 cm in width and 4.5 cm in length. With respect to the research focus, the ultraviolet protection factor (UPF) was computed for both the unblemished cotton and the other fabricated samples using the data recorded in accordance with Australia/New Zealand standard AC/NZS 439:1996. Equation (1) was employed to determine the UPF [27]:(1)$$UPF=\frac{\int_{280}^{400}\;E_{\lambda }\times S_{\lambda }\times d\lambda }{\int_{280}^{400}\;E_{\lambda }\times S_{\lambda }\times T_{\lambda }\times d\lambda }$$
where E is the relative erythemal spectral efficiency, S is the solar UV spectrum irradiance, T is the spectral transmittance of the specimen, dλ is the increase in wavelength (nm), and λ is the wavelength (nm).

#### 2.4.3. Scanning Electron Microscopy (SEM) and Transmission Electron Microscopy (TEM)

The elemental composition and surface morphology of the cotton fabric samples were captured employing a high-resolution SEM equipped with (Hitachi S-4800, Horiba equipment, Kyoto, Japan) running at 15 kV. TEM analysis (JEOL 2010; JEOL Ltd., Tokyo, Japan) was undertaken for observing the nano-sized activated carbon which applied high-energy electrons (300 KV) to accelerate to near the speed of light using an electron beam passing through a thin section specimen of material scatters electrons [28].

#### 2.4.4. Thermal Gravimetric Analysis

An instrument thermal gravimetric analysis (TGA 4000, PerkinElmer, Waltham, MA, USA) was used to conduct the thermal gravimetric analysis (TGA). The thermograms were taken in an air environment at a heating rate of 10 °C min^−1^ [29].

#### 2.4.5. Tensile Properties

Using a universal testing system with a constant rate of extension, the tensile characteristics of the untreated cotton and treated fabrics were evaluated. ASTM D5035-11 was strictly followed throughout the entire process. Clamping speed was held constant at 300 mm/min, and the gauge span (the clamps’ starting distance apart) was set to 75 mm [30].

#### 2.4.6. Air Permeability

An equipment manufactured by P.S.I Sales Pvt Ltd., India which calibrated to BS 3424, was used to measure the air penetrability of both untreated cotton and the treated cloth. The fabric sample size for this investigation was 5.07 cm^2^. A steady 1 cm head of water pressure decrease was used to calculate the air flow rate [31].

#### 2.4.7. Toughness Testing

The fabric’s toughness was evaluated by measuring its bending length using a Shirley Stiffness tester (V-5 Stiffness Tester, TABER, North Tonawanda, NY, USA). The specimen was kept at a size of 6″ × 1″ in accordance with the requirements of standard B.S. 3356:1961 [32].

#### 2.4.8. Wash Fastness Test

The FANC-coated cotton fabric specimen, 5 cm by 15 cm together with a white (bleached) cotton fabric, was agitated in a Launder ‘O’ meter (M228AA, SDL Atlas, Rock Hill, SC, USA) following the standard method AATCC 61-2A colorfastness to washing: accelerated laundering. The sample is then dried and assessed for both the color loss of the specimen and the staining of the adjacent fabric using grey scales. The color change of the specimen was assessed with a grey scale rating from 1 to 5 with the greatest change to no change, respectively, whereas the grey scale for the staining rates of a white cotton fabric was tested by a comparison with the specimen, where 1 indicates the greatest color transfer and 5 indicates no color transfer [33,34].

#### 2.4.9. Rubbing Test (Dry and Wet)

Plain woven white cotton fabric was mounted on the movable clam of the clock meter with 50 × 130 mm FANC-treated cotton fabric on the bed of the rubbing tester (M-204693, SDL Atlas, Rock Hill, SC, USA). The standard testing method was followed as per the procedure AATCC 8 for measuring the color fastness to rubbing. The rubbing test was carried out using the moveable clamp for 10 cycles per turn. The evaluation of the staining of the rubbing cloths (dry and wet conditions) were assessed using the grey scale for staining [33,34].

## 3. Results and Discussion

### 3.1. Characteristics of ANC and Developed FANCPCF

The grounded sample of Teff-hay-derived biocarbon was subjected to an XRD pattern examination. The pattern showed a turbostratic structure (Figure 3a). The resulting diffractograms exhibited the ratio of an area under two characteristic peaks due to graphitic crystallites (2θ = 6.3° and 2θ = 26.2°). It was apparent that the prepared ANC sample maintained its crystallinity after ball milling operation. In addition, to confirm the size measurements of ANC particles, the samples were undertaken to perform a particle size distribution using a laser beam scattering technique with sonication [35]. According to this result, the volume-based particle size distribution for the sample obtained is illustrated in Figure 3b. Based on this histogram, the average particle size fell below 100 nm. Figure 4 provides the SEM morphology of the ANC particles. As seen in Figure 4, the morphology of the activated carbon was understood to be uneven shapes and sizes. However, no clear pores were observed on the nano-structured carbon. Furthermore, the nano-sized particles were captured by the TEM analysis, and the image was presented in Figure 5. This depicts the transmission electron microscopy image of the synthesized carbon nanoparticles at a magnification of 20,000×. The observed characteristics revealed irregular shapes and an aggregated morphology of the synthesized carbon nanoparticles. These findings align with a previous study conducted by Varghese et al., 2013, where they investigated the carbon nanoparticles derived from kitchen soot and reported similar morphological properties [36].

Initially, by adopting the OVAT approach, the hybrid cotton materials were prepared based on the gradual increment in FANC (0, 1, 2, 3, and 4% by wt. of cotton) by keeping the concentration of an ANI constant (35 by wt.% of cotton) (Figure 6). In this approach, while testing the UV shielding property of each hybrid material, the sample contained FANC-3% and polyaniline-35% which was coded as FC_3_PA_35_, exhibiting the highest UV protection (UPF value of 57.9). Furthermore, the aniline wt.% was varied (10, 15, 20, 20, 25, and 30%) by keeping the concentration of FANC as 3 wt.% (Figure 7). In this approach, a cotton composite coated with 25 wt.% of aniline and 3 wt.% of FANC (coded as FC_3_PA_25_) showed the highest value of UPF (64.1). With this OVAT approach, it was apparent that the requirement approximate wt.% for FANC and ANI to coat on the cotton could be 3 and 25 wt.%, respectively. Consequently, additional optimization was examined to determine the optimal wt.% and interacting effect on these two limiting factors, FANC and ANI. In this line, the range limit for FANC and ANI were considered to be 2–4% and 20–30%, respectively.

### 3.2. Interaction Effect of ANI and FANC on UPF Value for Hybrid Nanocomposite

Based on the CCD design, FANCPCF samples were prepared using different ratios and the UPF value of the respective FANCPCF sample is presented in Table 1. In this line, the second-order-polynomial Equation (2) was developed in a way that correlates with the UPF value of the FANCPCF with ANI and FANC. The significance of the developed model was tested using an analysis of variance (ANOVA) that was found to be well satisfactory with the statistical acceptance (Table 2). Furthermore, the interaction effect for the UV shielding behavior in terms of UPF was generated as 3D response surface. The developed plots are presented in Figure 8a,b, as a response surface and contour plot, respectively. Figure 8c depicts the perturbation curves showing the individual effect of the composition of ANI and FANC against UPF which exhibited a significant deviation between the selected limits. Hence, it is confirmed that the factors were more influenceable for the UV protection. In addition, as seen in Figure 8d, the experimental and predicted the UPF values between the selected limits coincided. The analysis demonstrated that the highest UV shielding behavior of the FANCPCF increased with both ANI and FANC until reaching an optimal value. Furthermore, beyond this optimal composition, the nanocarbon-coated sample showed a reverse trend, whose UPF was found to be decreased [37]. The UPF drop may be traced back to the clustering and uneven dispersal of hybrids over the cotton material. The collected data suggest that a coating with an optimal ANC and FACN concentration could be ideal to achieve the highest UV protection. From the above results, the enhanced UV shielding tendency may be attributed to the synergistic effect contributed by both FANC and polyaniline hybrid materials [38]. It was well known that the FANC possesses a high surface area due to its nano form. Furthermore, the ability of the carbon nano-material to absorb the photons from the light provides a strong protection against the UV rays for the host cotton fabric. The statistical optimization revealed that the highest UPF value 64.563 can be attained with use of 24.585% ANI and 3.307% FANC. Comparably, the obtained result was 17.32 times higher than that of pristine cotton.
(2)$$UPF=-207.13634+15.96601ANI+43.26984FANC+0.5645ANI\times FANC\;-0.35259{ANI}^{2}-8.84225{FANC}^{2}$$

### 3.3. Spectral Analysis

FTIR: The amine-terminated silane was used to modify ANC surfaces for greater compatibility [39]. Figure 9 displays the Fourier transform-infrared spectroscopy of ANC and the surface of FANC. The existence of a peak at 1650 cm^−1^ indicates that a silane coupling agent, a Si–O–Si network, was present. The 3-aminopropyltriethoxysilane -CH_2_-aliphatic group stretches asymmetrically at 3455 cm^−1^ and symmetrically at 3365 cm^−1^. It has been observed that the surface amino groups can influence the polyaniline development via interfacial characteristics that regulate chain deposition and adhesion [40]. At the outset of polymerization, the oxidant APS transformed the amino (NH_2_) groups on the surface of FANC into radicals (NH_2_). To generate the organic or inorganic hybridized coating on the cotton fiber, an oxidant was used to kick off polyaniline growth from the FANC amino group.

Figure 10 shows that the FTIR spectra of both untreated cotton and cotton fabric have been coated with polyaniline and grafted to the FAC. The inset of Figure 8 shows a primitive cotton fabric, which exhibits many peaks between 1300 and 1800 cm^−1^ due to the stretching vibrations of the carbon–oxygen and carbon=oxygen functional groups. Cotton fabric may have carboxylic acid (-COOH) groups on its surface, as shown by a little signal at 2832 cm^−1^. The cloth turns a dark green color after the coating procedure, indicating that polyaniline was successfully grafted onto the cotton surface. Figure 8 may offer supporting data similar to that of a previous study like Mondal et al. [41]. Studies show that the fabrics with polyaniline grafting and FACN coating have different FTIR spectra. The C=N stretching vibrations in the polyaniline’s quinonoid and benzenoid rings account for the 2135 cm^−1^ and 2050 cm^−1^ peaks, while the C–N stretching vibration was responsible for the 1875 cm^−1^ peak.

### 3.4. Analysis of UV Shielding

Figure 11 displays the UV–visible transmitting spectrum of untreated cotton and treated cotton fabric (FANCPCF). Figure 3 show that, even after being bleached, the UPF of pristine cotton is rather low. Coatings made from a mixture of polyaniline polymer and ANC produced from biomass were tested, with the goal of improving the latter’s UV-blocking properties. The FANCPCF sample has a UPF rating of 64.5, making it the most UV-protective of the options considered here.

These findings suggest that the superior UV-blocking abilities of FANC and polyaniline hybrid materials are due to the synergistic effect provided by these two materials. It was widely recognized that the ANC’s nanoscale structure gave it a large surface area. The nanocarbon has the nanomaterial’s photon-absorption properties, further shielding the host cotton fabric from harmful ultraviolet radiation. The energy of carbon=carbon bonds was reported to be approximately 335 kJ, which is comparable to the energy of ultraviolet photons. Because of the ANC’s abundance of C=C bonds, which were spread in a graphitic order, the cellulose structure of fabrics was likely shielded from UV rays. The C=C structure of polyaniline affords the quininoid and benzenoid rings in its significant UV absorption behavior, which in turn protects against enhanced UV radiation. There is a synergistic effect that improves UV shielding performance when FANC and PANI are applied to the surface of cotton fabric.

In previous studies, the treatment of cotton with semiconducting metal oxide nanocomposites and nanocarbon has been shown to improve its UV-shielding properties (graphene and graphene oxide) [42]. Because of its organic nature, fabric would degrade under the photocatalytic action of semiconductor nanoparticles. The unwanted and convoluted method stems from the high cost and challenging reaction conditions inherent in the production of graphene compounds. Therefore, new UV-shielding materials that are more stable and absorb more UV light are in high demand. In accordance with this, hybridized cotton material using Teff-hay-derived FANC and polyaniline can perform towards potential results in terms of outstanding UV protection.

### 3.5. Morphological Analysis for FANCPCF

SEM was used to capture the morphological behavior of materials that made a substantial contribution to UV shielding. Figure 4 displays the morphological differences between fabrics with FANC particles and the hybrid cotton fabric with FANC coating grafted by polyaniline. The authors analyzed the features and compared them to those of 100% cotton. The smooth fiber structure of unbleached cotton is seen in Figure 12a. However, as shown in Figure 12b, the occurrence of FANC particles on the cellulose structure was indicated by the coarse surfaces with treated particles. The fact that the original fabric surface was not totally smooth was indicative of the sufficient interfacial bonding between FANC and fabric. The nanocarbon particles’ UV-blocking activity was further supported by evidence that they were intercalated between the warp and weft of the fibers. The nanorod-like structures seen all over the surface, represented in Figure 13b, indicated the occurrence of a PANI network. The network was discovered to be linear and flexible on a single dimension. It is important to remember that the optimized portion of FANC and polyaniline is responsible for the proper nucleation and growth to coat on the surface of the cotton textiles.

Thus, 139 GSM was calculated for FANCPCF, while 96 GSM was measured for the control fabric. The amino group of FANC can be polymerized with polyaniline via an interfacial process to provide a similar behavior, as previously indicated in FTIR. When the absorption of the FANC and ANI monomer was increased over a specific limit, polyaniline and FANC molecules can be intercalated and might be irregularly distributed across the fabric’s surface, generating agglomerates of polyaniline. Contrarily, the optimized FANCPCF displayed a more uniform FANC and polyaniline distribution. This proves that the optimal concentration of aniline and FAC is responsible for their superior UV shielding performance [43]. Another possible explanation was the silanization-achieved polyaniline layers that strongly interact with the consistently self-assembled 3APS-coated FANC. As a result, the homogeneous coating of polyaniline and FANC considerably improves the cotton fabric’s practical UV shielding performance. Figure 13a shows the elemental profile of pure cotton fabric and Figure 13b shows the profile of the optimized FANCPCF. It was found to have 65.9% nanocarbon atoms, while primitive cotton only contained 51.5%. Polyaniline builds up on the surface of the cotton, increasing the concentration of nanocarbons. Additionally, the presence of the nitrogen content (7.8%) in FANCPCF demonstrates the existence of polyaniline in the fabric’s construction.

### 3.6. Thermal Studies

Figure 14 shows the results of a comparison between the thermal degradation behaviors of primitive cotton and FANCPCF in air. Cotton fabric degrades in three phases, the last of which is connected with the loss of moisture at temperatures of approximately 100 °C.

The production of (aliphatic) char and volatilization are hallmarks of the initial stage of deterioration (300–400 °C). Significant deterioration has been seen in the second stage (400–600 °C) as a result of the carbonization process, which produces a wide variety of char residues (from aliphatic to aromatic). In other words, substantial combustion gases are produced as a result of the deterioration of crystallization of cellulose fiber. In the third and final stage of degradation (above 600 °C), carbonaceous species (CO and CO_2_) are completely decomposed or oxidized. Similar trends may be seen in the deterioration of FANCPCF samples. However, as shown in Table 3, their onset temperatures rise, suggesting that cotton coating and polyaniline grafting do, in fact, increase thermal stability to a small degree. Because of this increase in the thermal stability, the treated fabric is more desirable for use in high-end industrial applications requiring UV absorption [44]. In addition to a high char production, an efficient coating was also achieved.

### 3.7. Basic Characteristics of Primitive Cotton and FANCPCF

The results illustrate that coating and grafting the fabric specimen with FANC and polyaniline, respectively, result in improved UV shielding performance. However, coating and grafting are not supposed to have a major impact on the fabric’s fundamental qualities. Therefore, Table 4 displays the results of an examination of the fundamental characteristics of primitive cotton and FANCPCF, including their tensile strength, elongation, and stiffness. Air permeability testing was also performed to verify the existence of nanoparticles in the yarn’s voids. Tensile strength and elongation at breakage measurements yielded some intriguing findings. It was assumed that the acid condition used for grafting would cause the cotton textiles to lose their strong behavior after being grafted with polyaniline. Covering and fixing with FANC and PANI, however, improved the tensile strength and elongation in this study. Coated FANC, which preserves the original textile’s woven structure, the silane functionality over the FANC, and the interfacial process elucidated by FTIR and SEM, are possible causes of the increase in price. The PANI network formed over the FANC particles works as a force absorber because of these two events when subjected to external stress. These actions not only make textiles more flexible, but they also stop their structures from distorting. For this study, the bending length was utilized as a proxy for the stiffness attribute. The treated cloth was found to have a shorter bending length compared to the control fabric. Coating processes often boost the stiffness, and by extension, the bending length.

However, the bending length has not changed, demonstrating that the stiffness has not been excessively increased in this case. However, the following argument suggests that shorter bending lengths are not caused by a corresponding decrease in stiffness. Because the coated particles are nanoscale, they are able to penetrate the fabric’s interstices (the spaces between individual yarns or fibers) and deposit themselves there. This process adds weight to the cloth without compromising its strength. The shorter bending length is often attributed to an increase in fabric bulk. These data and analysis show that the tensile, elongation, and stiffness properties are unaffected by the coating and grafting procedures. According to the results, the air permeability of the hybrid coated textiles was found to be lower than that of the uncoated fabric. The existence of organic–inorganic hybrids on the cotton textiles’ surfaces was confirmed by a drop in their air permeability value. The decreased air permeability will not negatively affect the final use of the cloth because it is designed for industrial use. However, tensile strength is crucial in industrial applications, and this method actually boosts that property. Since the treatment did not compromise on the fundamental properties, it can be concluded that the UV resistance was greatly improved.

### 3.8. Durability Tests: Wash and Rubbing Fastness Tests

The wash fastness of the cotton fabric specimen coated and grafted with FANC and polyaniline showed a grey scale value between 4 and 5, indicating a very small or no color change in the specimen after washing. After 10 cycles of rubbing, the untreated white cotton fabric showed a value of 4–5 for dry rubbing and 4 for wet rubbing on the grey scales, showing that there was significantly less or approximately no transfer of color from the specimen to the white cotton fabric. The results were obtained from the strong bond attributed to the grafting of the activated nanocarbon with the cotton cellulose molecules using the optimized process condition. Additionally, the nano-sized FANC resulted in an excellent binding to the fabric because the high surface area can be a crucial factor for attaining a highly acceptable durability with respect to washing and rubbing. Figure 15a–e illustrate the developed nanocarbon-coated UV protective hybrid composite, primitive cotton fabric, the specimen fabric after the washing test, the adjacent fabric sample used for the washing test, the fabric used for dry rubbing test, and the fabric used for the wet rubbing test.

## 4. Conclusions

An affordable functionalized activated nano-structured carbon (FANC) was prepared from Teff hay biomass residues. Aiming to develop a cotton-based hybrid material with maximum possible UV protective properties, the FANC and polyaniline weight percent was optimized using the RSM-statistical approach. In this context, the improved UV-shielding behavior of the cotton fabric was determined have the highest UPF value of 64.563 that can be attained with the use of 24.585% ANI and 3.307% FANC. Comparably, the obtained result was 17.32 times higher than that of pristine cotton. The basic characteristics of the hybrid cotton material with an optimized condition was found to be a well-appreciable tensile strength, elongation at break, and bending length. Hence, increasing the fabric’s UV resistance by a treatment, this coating was successful since it does not reduce the fabric’s basic properties. The incorporation of organic hybrids that create a coating on the cloth and their resulting morphologies were also explored using SEM and EDX analysis. The transmission spectrum of the UV-visible range for the developed FANC-based hybrid cotton showed a potential ability of absorption for UV rays. Technocrats can put these treated cotton textiles to good use as UV protection textile materials since the organic–inorganic hybrids of FANC and polyaniline are inexpensive and ecologically benign. As a result, the produced material has an enhanced UV shielding capability, making it suitable for high-performance industrial applications.

## Figures and Tables

**Figure 1 molecules-28-05142-f001:**
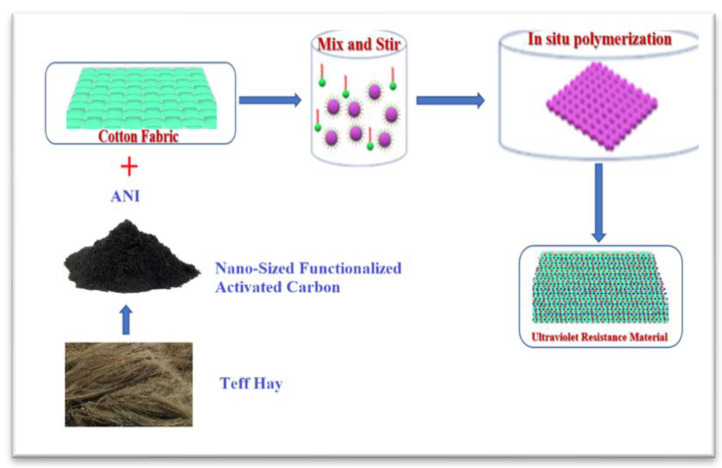
A schematic flow for the in situ polymerization of Teff hay-based bionanocarbon-doped polyaniline on cotton fabric.

**Figure 2 molecules-28-05142-f002:**
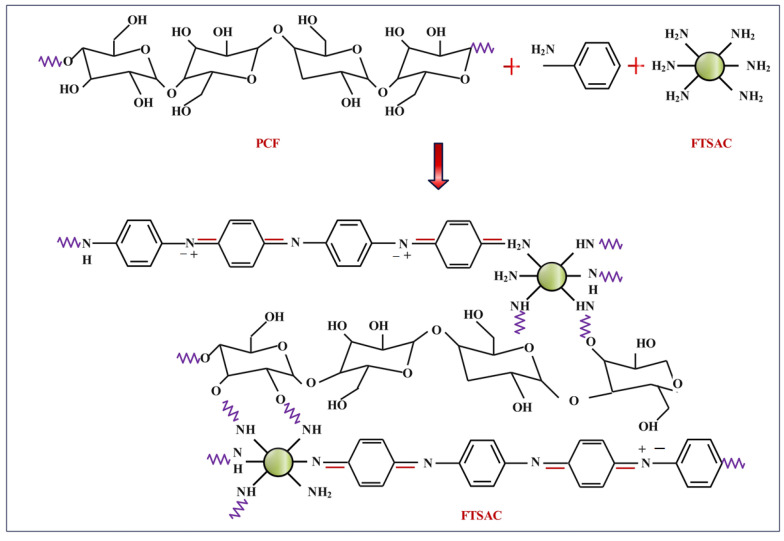
Reaction mechanism for grafting process whereby a hybrid nanocomposite was prepared.

**Figure 3 molecules-28-05142-f003:**
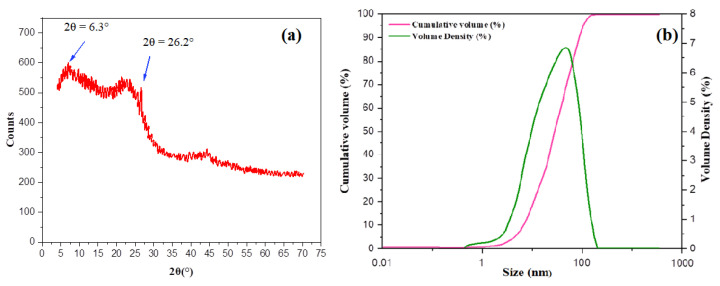
XRD pattern of Teff-hay-derived nanocarbon (**a**) and volume-based particle size distribution for the sample (**b**).

**Figure 4 molecules-28-05142-f004:**
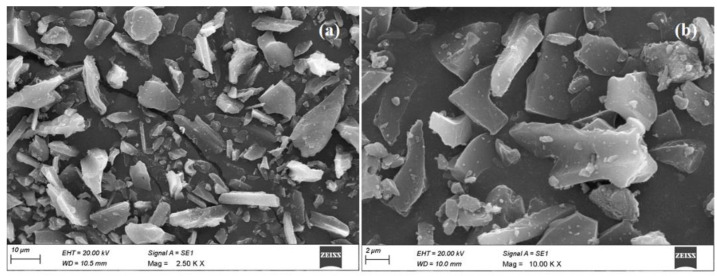
The SEM image of the Teff-hay-derived nanocarbon (**a**) for 2.5 K magnification and (**b**) 10.00 K magnification.

**Figure 5 molecules-28-05142-f005:**
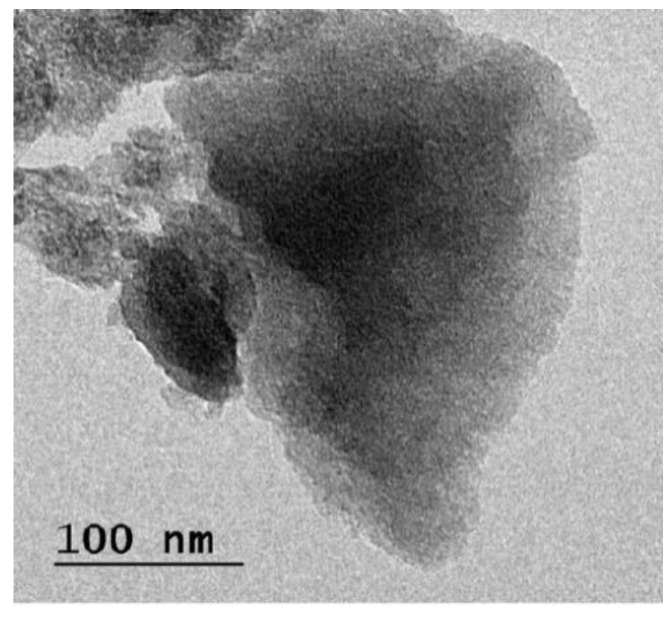
TEM image of the nano-sized carbon.

**Figure 6 molecules-28-05142-f006:**
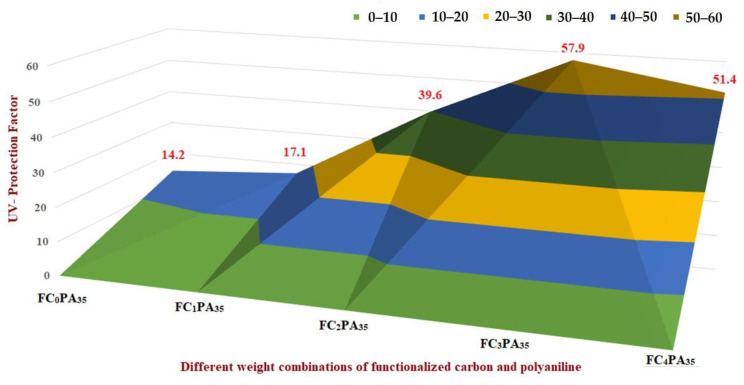
OVAT approach based on the gradual increment in FANC.

**Figure 7 molecules-28-05142-f007:**
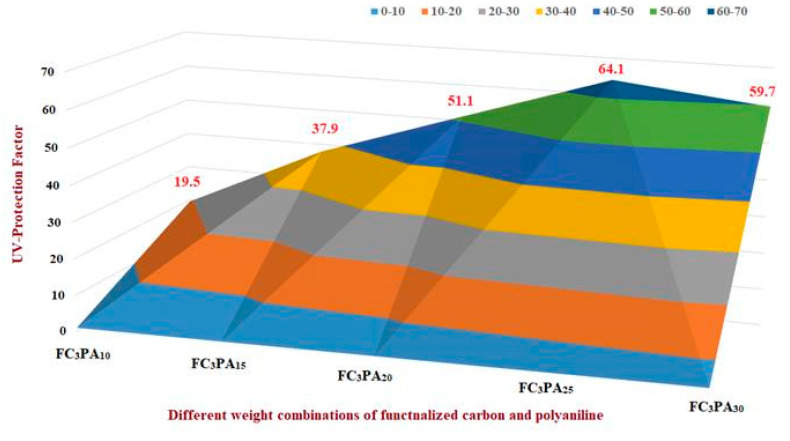
OVAT approach based on the gradual increment in PANI.

**Figure 8 molecules-28-05142-f008:**
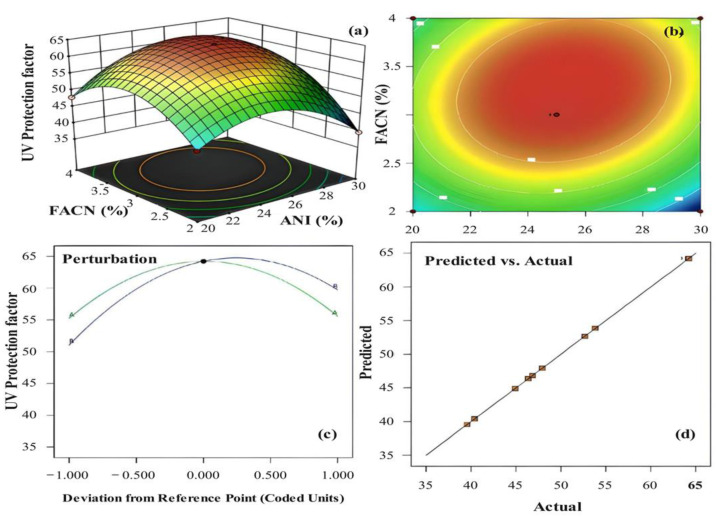
The developed RSM plot for the interaction effect between ANI and FANC on UPF (**a**), the contour plot (**b**), the perturbation plot (**c**), and predicted vs. actual plot (**d**).

**Figure 9 molecules-28-05142-f009:**
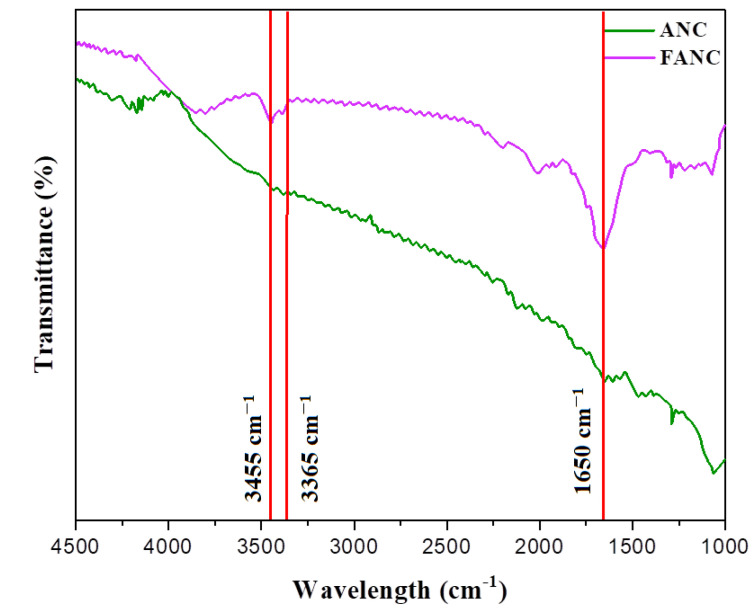
FTIR spectrum of ANC and FANC.

**Figure 10 molecules-28-05142-f010:**
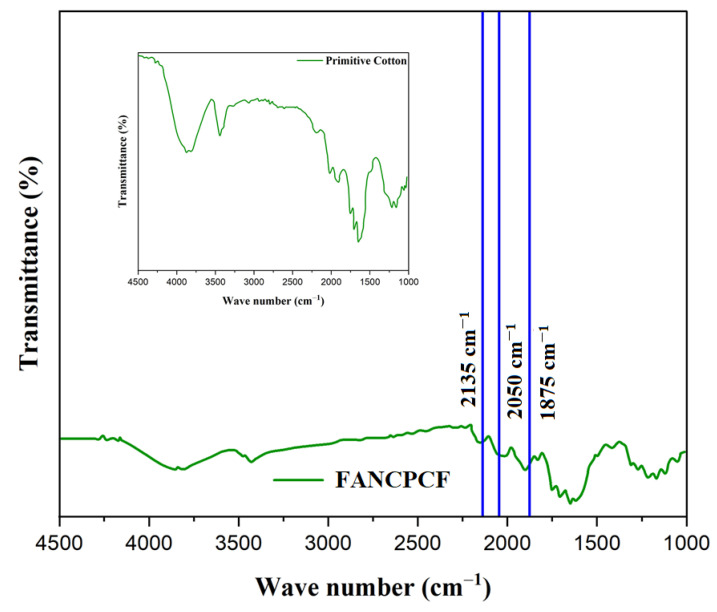
FTIR spectrum of primitive cotton (inset) and FANCPCF.

**Figure 11 molecules-28-05142-f011:**
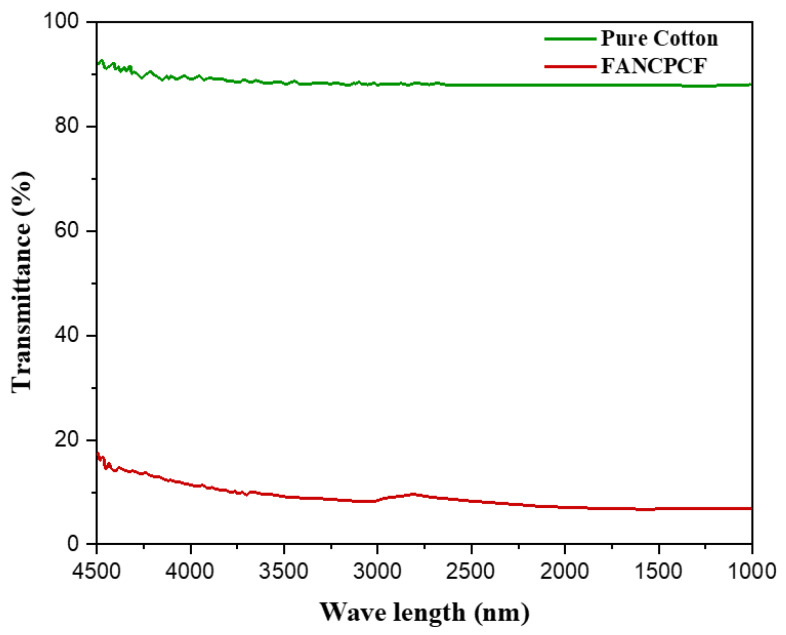
Transmission spectra in the UV–visible range for primitive cotton and FANCPCF.

**Figure 12 molecules-28-05142-f012:**
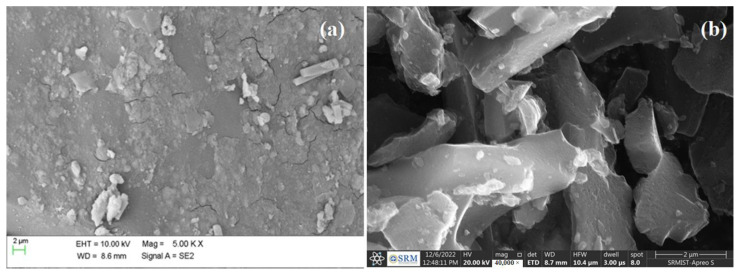
SEM morphological observation of (**a**) unbleached cotton and (**b**) FANCPCF.

**Figure 13 molecules-28-05142-f013:**
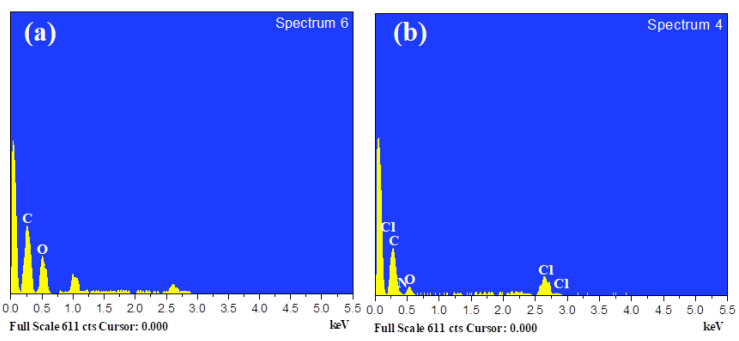
EDAX profiles of (**a**) primitive cotton and (**b**) FANCPCF.

**Figure 14 molecules-28-05142-f014:**
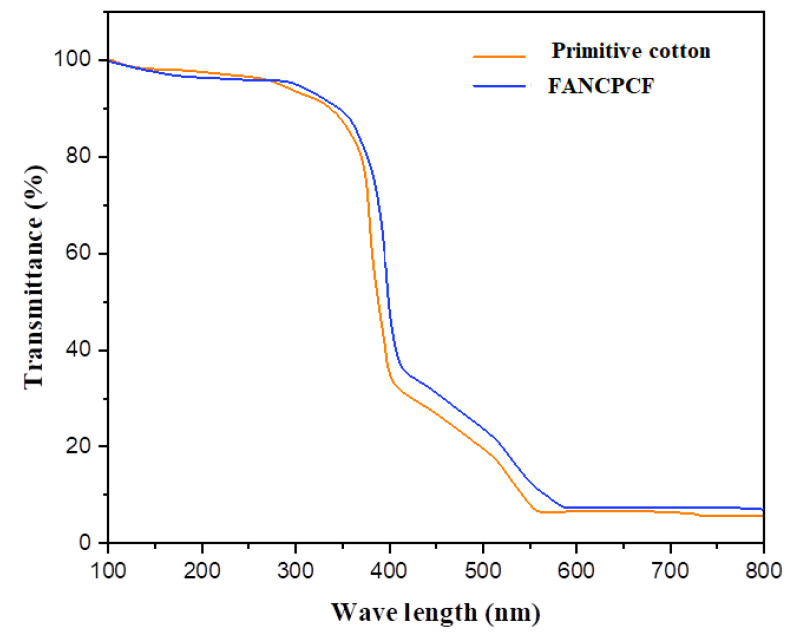
TGA profiles of primitive cotton and FANCPCF.

**Figure 15 molecules-28-05142-f015:**
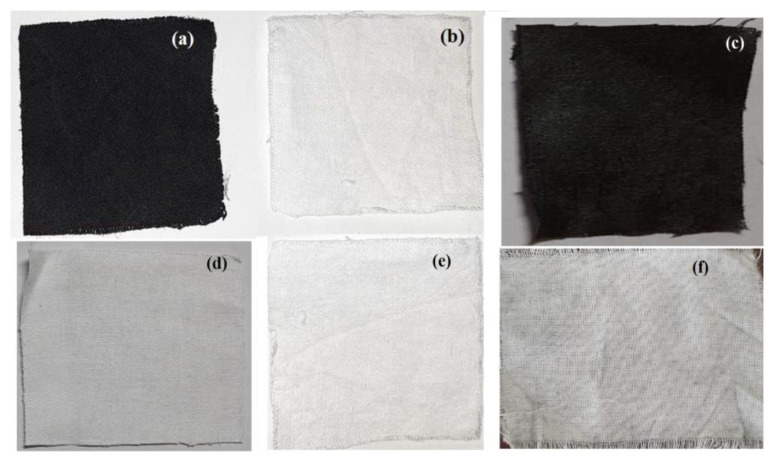
(**a**) The developed nanocarbon-coated UV protective hybrid composite, (**b**) primitive cotton fabric, (**c**) the specimen fabric after washing test, (**d**) adjacent fabric sample used for washing test, (**e**) the fabric used for dry rubbing test, and (**f**) the fabric used for wet rubbing test.

**Table 1 molecules-28-05142-t001:** The CCD experimental design for various combinations of ANI and FANC.

Sl. No	ANI (%)	FANC (%)	UPF
1	25	1.58579	40.39
2	30	4	53.82
3	32.0711	3	46.82
4	25	3	64.25
5	17.9289	3	46.36
6	20	2	44.92
7	25	3	64.2
8	25	3	64.2
9	25	3	64.21
10	25	4.41421	52.68
11	25	3	64.2
12	30	2	39.55
13	20	4	47.9

**Table 2 molecules-28-05142-t002:** The ANOVA statistical analysis of developed model (Equation (2)).

Source	Sum of Squares	df	Mean Square	F-Value	*p*-Value	
Model	1141.25	5	228.25	2.79 × 10^5^	<0.0001	Significant
A-PA	380.47	1	380.47	4.65 × 10^5^	<0.0001	
B-FC	162.1	1	162.1	1.98 × 10^5^	<0.0001	
AB	31.87	1	31.87	38,940.36	<0.0001	
A^2^	540.52	1	540.52	6.61 × 10^5^	<0.0001	
B^2^	543.9	1	543.9	6.65 × 10^5^	<0.0001	
Residual	0.0057	7	0.0008			
Lack of fit	0.0038	3	0.0013	2.73	0.1783	Not significant
Pure error	0.0019	4	0.0005			
Cor total	1141.26	12				

**Table 3 molecules-28-05142-t003:** Thermal characteristics of primitive cotton, and FANCPCF.

	Degradation Temperatures		
Samples	T_5%_ (°C)	T_10%_ (°C)	Char Yield at 750 °C
Primitive cotton	231	282	0
CA_25_Ca_3_	244	297	2.1

**Table 4 molecules-28-05142-t004:** Basic characteristics of the treated samples.

Sample	Tensile Strength (lbf)		Elongation at Break (%)		Bending Length (cm)		Air Permeability (cc/s/cm^2^)
	Weft	Warp	Weft	Warp	Weft	Warp	
Primitive cotton	68.1	74.2	28.6	26.1	1.97	2.06	283.7
FANCPCF	77.35	79.36	33.15	31.98	1.77	1.93	101.8

## Data Availability

The data presented in this study are available upon request from the corresponding author.

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
