# Peer review of "Fabrication of Enhanced UV Protective Cotton Fabric Using Activated Nano-Biocarbon Derived from Teff Hay Grafted by Polyaniline: RSM-Based Optimization and Characterization"

_molecules, 2023, doi:10.3390/molecules28135142_

Round 1

Reviewer 1 Report

This is an interesting, original and complete research manuscript comprising useful data, and I recommend the acceptance after minor revision according to the following comments.

1. Abstract is too long, please shortens and clearly highlights the important results.

2. Title is little primitive, please revise.

3. The SEM/XRD analysis must be studied.

4. Conclusion is just repetition of abstract?

5. Both language and technical stream of manuscript contents is good; however, typos must be polished.

6. The city and country for each instrument in experimental section must be identified.

7. Please give the abbreviations of every short form at first used.

8. The standard methods/procedures must be clearly identified in the experimental part.

Both language and technical stream of manuscript contents is good; however, typos must be polished.

Author Response

Reviewer 1: (All the response have been incorporated in the revised manuscript the same has been highlighted in Yellow color)

Reviewer 2 Report

The manuscript is about polymerization of Teff hay-based PANI-grafted nano biocarbon on cotton fabric to improve its UV shielding properties. The detailed comments are as follows:

Section 2: Material and Methods

(a)   Please state the fabric type (e.g. plain, twill, sateen or etc.,) used in this study.

Section 3: Results and discussion

(a)   With regard to Figure 3, please state the magnification used for TEM observation and make discussion on the finding(s), such as the morphology of the carbon, according to the TEM image.

Insufficient test

(a)   It is highly recommended that the authors should add tests concerning the durability of the coated material on cotton fabric such as washing test and abrasion resistance.

Typos

(a)   Certain typos are found, such as line 132: concertation; line 139: Figigure.1. It is strongly suggested that the authors should check the English spelling carefully.

Author Response

Reviewer 2: (All the response have been incorporated in the revised manuscript the same has been highlighted in Turquoise color)

Round 2

Reviewer 1 Report

none

Author Response

Since there was no comment, response is not applicable.

Reviewer 2 Report

The manuscript is about polymerization of Teff hay-based PANI-grafted nano biocarbon on cotton fabric to improve its UV shielding properties. The detailed comments are as follows:

Section 2: Materials and methods

(a)     It is highly recommended that the authors should add a subsection (i.e., Section 2.4) to introduce the characterization techniques and durability test (from FTIR to rubbing test) instead of integrating it into Section 2.3 which introduces about fabrication.

(b)     For the washing and rubbing test, please state the test name, test number, and the standard test organization, for example, AATCC test method 61 Colorfastness to washing: accelerated laundering.

Section 3.8: Durability tests: wash and rubbing fastness tests

(a)     Please provide images of the samples before and after the washing and rubbing fastness tests.

(b)     Please provide the rating of the tested samples according to the standard test procedure with the use of gray scale.

Reference list

(a)     Please check the references in the reference list carefully. Two references with the same reference no. [34] and [35] are found. Please rearrange and check all the references in the reference list carefully. In addition, please also check whether there is irrelevant or inappropriate reference cited.

Author Response

Reviewer 2: (All the responses have been incorporated in the revised manuscript the same has been highlighted in Yellow color)

Section 2: Materials and methods

(a)     It is highly recommended that the authors should add a subsection (i.e., Section 2.4) to introduce the characterization techniques and durability test (from FTIR to rubbing test) instead of integrating it into Section 2.3 which introduces about fabrication.

Response: In the revised manuscript, the comment has been addressed; the same has been highlighted in the yellow color. 

(b)     For the washing and rubbing test, please state the test name, test number, and the standard test organization, for example, AATCC test method 61 Colorfastness to washing: accelerated laundering.

Response: In the revised manuscript, the comment has been addressed; the same has been highlighted in the yellow color. 

 Section 3.8: Durability tests: wash and rubbing fastness tests

(a)     Please provide images of the samples before and after the washing and rubbing fastness tests.

Response: In the revised manuscript, the comment has been addressed; the images have been provided accordingly.   

(b)     Please provide the rating of the tested samples according to the standard test procedure with the use of gray scale.

 Response: In the revised manuscript, the comment has been addressed in the section for Durability tests: wash and rubbing fastness tests; the same has been highlighted in the yellow color. 

Reference list

(a)     Please check the references in the reference list carefully. Two references with the same reference no. [34] and [35] are found. Please rearrange and check all the references in the reference list carefully. In addition, please also check whether there is irrelevant or inappropriate reference cited.

Response: In the revised manuscript, the comment has been addressed in the references list; the same has been highlighted in the yellow color. 

Round 3

Reviewer 2 Report

The revised manuscript is now acceptable for publication.